# Sph2_(176–191)_ and Sph2_(446–459)_: Identification of B-Cell Linear Epitopes in Sphingomyelinase 2 (Sph2), Naturally Recognized by Patients Infected by Pathogenic Leptospires

**DOI:** 10.3390/vaccines11020359

**Published:** 2023-02-04

**Authors:** Laura Sant’Anna Ataides, Fernanda de Moraes Maia, Fernando Paiva Conte, Lourdes Isaac, Angela Silva Barbosa, Josué da Costa Lima-Junior, Kátia Eliane Santos Avelar, Rodrigo Nunes Rodrigues-da-Silva

**Affiliations:** 1Laboratório de Tecnologia Imunológica, Instituto de Tecnologia em Imunobiológicos, FIOCRUZ, Rio de Janeiro 21040-900, RJ, Brazil; 2Laboratório Piloto Eucariotos, Instituto de Tecnologia em Imunobiológicos, FIOCRUZ, Rio de Janeiro 21040-900, RJ, Brazil; 3Departamento de Imunologia, Instituto de Ciências Biomédicas, Universidade de São Paulo, São Paulo 05508-000, SP, Brazil; 4Laboratório de Bacteriologia, Instituto Butantan, São Paulo 05503-900, SP, Brazil; 5Laboratório de Imunoparasitologia, Instituto Oswaldo Cruz, FIOCRUZ, Rio de Janeiro 21040-900, RJ, Brazil; 6Laboratório de Referência Nacional para Leptospirose, Instituto Oswaldo Cruz, FIOCRUZ, Rio de Janeiro 21040-900, RJ, Brazil

**Keywords:** leptospirosis, B-cell epitope, immunoinformatic

## Abstract

Sphingomyelin is a major constituent of eukaryotic cell membranes, and if degraded by bacteria sphingomyelinases may contribute to the pathogenesis of infection. Among *Leptospira* spp., there are five sphingomyelinases exclusively expressed by pathogenic leptospires, in which Sph2 is expressed during natural infections, cytotoxic, and implicated in the leptospirosis hemorrhagic complications. Considering this and the lack of information about associations between Sph2 and leptospirosis severity, we use a combination of immunoinformatics approaches to identify its B-cell epitopes, evaluate their reactivity against samples from leptospirosis patients, and investigate the role of antibodies anti-Sph2 in protection against severe leptospirosis. Two B-cell epitopes, Sph2_(176-191)_ and Sph2_(446-459)_, were predicted in Sph2 from *L. interrogans* serovar Lai, presenting different levels of identity when compared with other pathogenic leptospires. These epitopes were recognized by about 40% of studied patients with a prevalence of IgG antibodies against both Sph2_(176-191)_ and Sph2_(446-459)_. Remarkably, just individuals with low reactivity to Sph2_(176-191)_ presented clinical complications, while high responders had only mild symptoms. Therefore, we identified two B-cell linear epitopes, recognized by antibodies of patients with leptospirosis, that could be further explored in the development of multi-epitope vaccines against leptospirosis.

## 1. Introduction

Leptospirosis is a tropical and neglected emerging zoonotic disease that afflicts humans and other animals [1]. It is considered a global public health problem, with an estimated one million new leptospirosis cases reported annually and a mortality rate of about 60,000 [2]. Although the disease occurs worldwide, it is most prevalent in tropical countries, where conditions for transmission are most favorable [3]. In Brazil, leptospirosis is a serious public health problem because over 3500 cases are reported annually, leading to an up to 75% hospitalization rate and resulting in the death of about 10% of patients [4,5].

The disease is caused by pathogenic spirochetes from the genus *Leptospira*, which can infect humans and almost all mammals, as well as reptiles and amphibians [6]. The transmission occurs when bacteria from contaminated soil or water come into contact with cutaneous lacerations or mucous membranes of the mammalian host, and disseminates via the bloodstream to many organs [7]. Leptospirosis clinical manifestations in humans present a broad spectrum of symptoms that are often mistaken for other infections, varying from mild fever, in the acute phase, to Weil’s syndrome, characterized by multiorgan failure that frequently leads to death [8]. Among its severe outcomes, pulmonary hemorrhagic syndrome is a feared complication caused by damage to the endothelial lining of blood vessels, which can be fatal in approximately 50% of cases [9].

Despite the impact of leptospirosis on public and animal health, there is a lack of effective vaccines against the more than 300 antigenically diverse serovars of pathogenic *Leptospira* [1,10]. Until now, vaccines available against leptospirosis were either based on inactivated bacteria or membrane preparations from pathogenic *Leptospira* species, which do not provide cross-protection among the pathogenic *Leptospira* species and are associated with severe side effects [11]. Hence, molecules associated with pathogenic *Leptospira* spp. pathogenesis, virulence, infectivity, and survival have been investigated as both therapeutic targets and vaccine candidates.

Several virulent factors produced by pathogenic leptospires are associated with the diverse damage caused by the bacterium in humans [12]. Among these virulent factors, sphingomyelinases are implicated in leptospirosis hemorrhagic complications due to their activity in host cell membranes, by catalyzing the hydrolysis of sphingomyelin. Moreover, these molecules are potentially involved in immune evasion and nutrient acquisition [13,14]. Corroborating their importance, sphingomyelinases produced by *Staphylococcus aureus* and *Listeria ivanovii* are directly associated with their infectivity in animal models [15,16], suggesting that this kind of molecule could be investigated as a vaccine candidate or therapeutic target.

There are five sphingomyelinases (Sph1, Sph2, Sph3, Sph4, and SphH) produced by pathogenic leptospires that are absent in the nonpathogenic *L. biflexa* [17]. Among these five proteins, Sph2 is highlighted by its structural similarity to SmcL, the sphingomyelinase from *L. ivanovii*, by being a Mg^++^ dependent hemolysin with demonstrated sphingomyelinase and hemolytic activities [18], and by having a recognized ability to damage lymphocytes and macrophages [19], hypothetically helping leptospiral defense against the host immune system. On another hand, despite the presence of anti-Sph2 antibodies in patients [20] and in mares that suffered abortions due to this infection [21], and the knowledge of the metal-binding site and catalytic site of Sph2 [18], its B-cell epitopes and the associations of anti-Sph2 with protection against leptospirosis complications remain unexplored.

Therefore, this was the first study aiming to predict linear B-cell epitopes in *L. interrogans* Sph2, to evaluate their recognition as synthetic peptides antigens by serum samples from leptospirosis patients, and to investigate the associations between the specific immune responses to each validated epitope and the clinical data of patients. This strategy has been used successfully to identify epitopes for the development of novel vaccines, diagnostic methods, and therapeutics [22,23,24,25,26,27,28].

## 2. Materials and Methods

### 2.1. Studied Proteins

To predict B-cell linear epitopes and perform in silico analyses, we used the entire sequence of Sph2 of *L. interrogans* serogroup *Icterohaemorrhagiae* and serovar *Lai* (Uniprot ID: P59116), obtained from the Uniprot database https://www.uniprot.org/ (accessed on 24 March 2020), as the sequence reference to this study. Moreover, to evaluate the similarity of Sph2 among pathogenic leptospires, we compared the used sequence with the described sequences of Sph2 from *L. interrogans* (serogroups: Australis, Bataviae, and Pyrogenes); *L. alexanderi* (serogroup: Manhao); *L. alstonii* (serogroup: Ranarum); *L. borgpetersenii* (serogroups: Pomona and Sejroe); *L. noguchii* (serogroups: Autumnalis and Panama); *L. kirschneri* (serogroup: Autumnalis); *L. santarosai* (serogroup: Javanica); and *L. weilii* (serogroups: Ranarum and Tarassovi). In addition, to investigate the conservation of reference Sph2 and other *Leptospira* spp. sphingomyelinases, this sequence was also aligned and compared with sequences of SphH (Uniprot ID: O34095), Sph1 (Uniprot ID: P59115), Sph3 (Uniprot ID: A0A0E2DC81), and Sph4 (Uniprot ID: A0A0E2DCF7) from *L. interrogans* serogroup Icterohaemorrhagiae and serovar Lai. Finally, the Sph2 reference sequence was also compared with sphingomyelinases from *Listeria ivanovii* (Uniprot ID: Q9RLV9), *Bacillus cereus* (Uniprot ID: P09599), *Staphylococcus aureus* (Uniprot ID: A0A7U4AUV1), and *Pseudomonas sp.* (Uniprot ID: Q93HR5). All studied sequences were accessed on 24 March 2020 and are summarized in Table 1.

### 2.2. Epitope Prediction

To predict B-cell linear epitopes, the amino acid sequence of Sph2 (Uniprot ID: P59116) was evaluated by a combination of 9 algorithms: Bepipred and Emini Surface Accessibility prediction (ESA), from Immune Epitope Database (IEDB: http://tools.iedb.org/bcell/ (accessed on 10 June 2020); ABCpred http://crdd.osdd.net/raghava/abcpred/ (accessed on 10 June 2020); ElliPro http://tools.iedb.org/ellipro/ (accessed on 10 June 2020); BCePred http://crdd.osdd.net/raghava/bcepred/ (accessed on 10 June 2020) to evaluate the exposed residues scale (ERS) [29], polarity scale (PS) [30], and hydrophilicity scale (HS) [31]; LBtope https://webs.iiitd.edu.in/raghava/lbtope/index.php (accessed on 10 June 2020); and COBEpro http://scratch.proteomics.ics.uci.edu (accessed on 10 June 2020).

Regarding the used algorithms, Bepipred combines the hidden Markov model with the propensity scale by Parker et al. to predict linear B-cell epitopes through the sequence of the protein in FASTA format [32]. Emini Surface Accessibility prediction indicates the probability of a peptide being found on the surface of a protein by calculating the surface accessibility of hexapeptides [33]. ABCpred is a server that allows the prediction of continuous B-cell epitopes in a protein through the amino acid sequence with 65.93% accuracy according to its validation tests [34]. ElliPro is a web tool that allows the prediction of antibody epitopes in a protein amino acid sequence or structure with the implementation of a modified version of Thornton’s method, residue clustering algorithm, and the MODELLER program [35]. BCePred allows users to predict B-cell epitopes using physicochemical properties (hydrophilicity, flexibility/mobility, accessibility, polarity, exposed surface, and turns) or a combination of properties [36]. LBtope is a web server that predicts linear B-cell epitopes through SVM-based models using dipeptide composition generated from the query sequence(s). The overall accuracy of this server is approximately 81% [37]. COBEpro is a two-step system for predicting continuous B-cell epitopes, that first uses a support vector machine to make predictions on short peptide fragments within the query antigen sequence and then calculates an epitopic propensity score for each residue based on the fragment predictions. Secondary structure and solvent accessibility information (either predicted or exact) can be incorporated to improve performance. COBEpro is incorporated into the SCRATCH prediction suite at http://scratch.proteomics.ics.uci.edu [38]. All the mentioned algorithms were used considering their default thresholds. Sequences with at least 10 amino acids, predicted by at least five algorithms, were considered predicted B-cell linear epitopes.

### 2.3. Antigenicity Analysis

Each predicted epitope was evaluated for antigenicity by the VaxiJen algorithm http://www.ddg-pharmfac.net/vaxijen/VaxiJen/VaxiJen.html (accessed on 12 June 2020) with the default threshold (0.4). VaxiJen is a protective antigen prediction server that allows classification based only on the physical–chemical properties of the protein of interest [39].

### 2.4. Conservation Analysis

Sequence alignment and identification of conserved patterns among sphingomyelinases from *L. interrogans* serovar Lai [Sph2 (Uniprot ID: P59116), Sph1 (Uniprot ID: P59115), Sph3 (Uniprot ID: A0A0E2DC81), Sph4 (Uniprot ID: A0A0E2DCF7), and SphH (Uniprot ID: O34095)] and other bacteria including *Listeria ivanovii* (SmcLUniprot ID: Q9RLV9), *Bacillus cereus* (BC SMase, Uniprot ID: P09599), *Staphylococcus aureus* (Uniprot ID: A0A7U4AUV1), and *Pseudomonas* spp. strain TK4 (Uniprot ID: Q93HR5) were conducted by MAFFT [40], using the software MegAlign pro. In the same way, to verify the conservation degree of Sph2 among pathogenic Leptospira spp., the reference Sph2 was aligned and compared with all other 17 Sph2 described in Table 1. To investigate the conservation degree of predicted epitopes among pathogenic leptospires, each predicted sequence was aligned and compared with all Sph2 sequences described in Table 1. Values of identity (%) represent the percentage of equal amino acids aligned. Moreover, we compared the conservation of amino acids present in the catalytic sites, and metal-binding sites were compared among Sph2 from pathogenic leptospires and other sphingomyelinases, based on the study of Narayanavari S.A. and collaborators [18].

### 2.5. Peptide Synthesis

Sequences predicted as antigenic linear B-cell epitopes were synthesized using fluorenylmethoxycarbonyl (F-moc) solid-phase chemistry [41,42] (WatsonBio, Houston, TX, USA). Analytical chromatography of the peptides demonstrated a purity degree higher than 95%, and mass spectrometry analysis of the peptides indicated estimated masses corresponding to the molecular masses of the peptides.

### 2.6. Studied Population

In this study, 87 serum samples of Brazilian patients were provided by the National Reference Laboratory for Leptospirosis — Fiocruz-RJ. All samples were previously tested by microscopic agglutination test (MAT), resulting in 51 leptospirosis patients (MAT positive), reactive in MAT, and 36 negative controls (MAT negative). The study was reviewed and approved by the Oswaldo Cruz Foundation Ethical Committee and the National Ethical Committee of Brazil (number CAAE: 31405820.8.0000.5262).

### 2.7. Evaluation of Natural Immunogenicity of Predicted Epitopes

Samples of confirmed leptospirosis cases and the control group were screened for the presence of naturally acquired antibodies against the synthetic peptides via ELISA as previously described [24].

Briefly, MaxiSorp 96-well plates (Nunc, Rochester, NY, USA) were coated with 100 μg/mL of a peptide. After overnight incubation at 4 °C, plates were washed with phosphate-buffered saline (PBS) and blocked with PBS-containing 5% non-fat dry milk (PBS-M) for 1 h at 37 °C. Individual serum samples diluted 1:100 on PBS-M were added in duplicate wells, and the plates were incubated at 37 °C for 1.5 h. After three washes with PBS-Tween20 (0.05%), bound antibodies were detected with peroxidase-conjugated goat anti-human IgG (SouthernBiotech, catalog number: 2048-05) or goat anti-human IgM (catalog number: 2020-05), diluted at 1:1000 (in PBS-M), and incubated for 1 h at 37 °C, followed by TMB (3,3′,5,5′-tetramethylbenzidine). The reaction was stopped by the addition of HCl (1N), and the absorbance was read at 450 nm using an xMark™ microplate absorbance spectrophotometer (Bio-Rad, Hercules, CA, USA). The results for total IgG and IgM were expressed as the reactivity index (RI)—the ratio between the mean optical density (OD) of tested samples and the mean OD of 44 control group samples plus 2.5 standard deviations (SD). Subjects were considered IgG responders to a particular antigen if the RI was higher than 1.

### 2.8. Statistical Analysis of Data

The obtained data were analyzed using GraphPad Prism 8.0 (GraphPad Software, Inc., San Diego, CA, USA). First, to determine if a variable was normally distributed, the one-sample Kolmogorov–Smirnoff test was used. Differences in frequencies of IgG and IgM responders to synthetic peptides were evaluated using Fisher’s exact test, while the reactivity indices against synthetic peptides between responders to each epitope were compared using the Mann–Whitney test. A two-sided *p*-value < 0.05 was considered significant.

## 3. Results

### 3.1. Predicted Epitopes

In this study, we used a combination of nine algorithms to predict linear B-cell epitopes from *L. interrogans* serovar Lai Sph2. In this way, considering that each amino acid residue was predicted as inserted in a linear B-cell epitope by at least five algorithms, we identified two sequences as continuous B-cell epitopes in the studied protein: GHDERAKRISKSDYVK (Sph2_(176-191)_) and TPTKSGHKKKYDQV (Sph2_(446-459)_). As shown in Table 2, both sequences were entirely or partially predicted by each used algorithm.

### 3.2. Antigenicity Assessment

The protein Sph2 was considered a protective antigen, obtaining a score of 10.458 in the VaxiJen algorithm. Regarding predicted B-cell linear epitopes, both sequences, Sph2_(176-191)_ and Sph2_(446-459)_, were predicted as antigenic, presenting Vaxijen scores of 1.173 and 1.763, respectively.

### 3.3. Conservation Analysis

First, as indicated in Table 3, Sph2 is a conserved protein among pathogenic leptospires, with a mean identity of 69.2%, which ranges from 47.2% to 100% when compared with Sph2 of pathogenic leptospires, and ranges from 53% (SphH) to 99.3% (Sph4) when compared with other *Leptospira* spp. sphingomyelinases. Moreover, when compared with sphingomyelinases of other bacteria, Sph2 presented identities ranging from 42% to 49% (Table 3).

Regarding the conservation degree of predicted epitopes, the epitopes Sph2_(176-191)_ and Sph2_(446-459)_ were considered non-conserved among microorganisms not belonging to the genus *Leptospira*, because they presented an *E-value* greater than one on BlastP. Interestingly, compared with Sph2 from other pathogenic leptospires and other sphingomyelinases, we observed a different profile of conservation between both epitopes. Sph2_(176-191)_ presented more than 70% of identity with pathogenic leptospires, highlighting 100% of identity in 20% of studied pathogenic leptospires (*L. interrogans* serovar Bataviae and *L. noguchii* serovars Panama and Autumnalis). This epitope also presented 100% of identity when compared with *L. interrogans* serovar Lai Sph4, and presented a mean identity of 42.2% when compared with sphingomyelinases from other bacteria. On the other hand, Sph2_(446-459)_ was highly conserved among all studied leptospires SPh2, presenting more than 70% of identity. Interestingly, the epitope Sph2_(446-459)_ was also highly conserved in other *Leptospira interrogans* serovar Lai sphingomyelinases, presenting identities that ranged from 71.4% to 100%, but presented low conservation when compared with other studied bacteria sphingomyelinases, presenting identities less than 37.5% (Table 3).

### 3.4. Epitopes Location and Sph2 Active Sites

In 2012, Narayanavari and collaborators described the main amino acids that compose the SPh2 catalytic and metal-binding sites [18]. Based on their study, we investigated the conservation of both catalytic and metal-bind sites among SPh2 of studied pathogenic leptospires. As shown in Table 4, there are four main amino acids described in the *L. interrogans* serovar Lai Sph2 catalytic site (H293, D393, Y394, and H433) and five in the central metal-binding site (N161, E200, D341, N343, and D432). Considering this, when we compared aligned amino acids in Sph2 from *L. interrogans* serovar Lai with other pathogenic leptospires, the histidine at position 293 (H293) was the unique amino acid conserved in all studied leptospires. Moreover, the number of modified amino acids in the catalytic or metal-binding sites ranged from one to six among the studied Sph2. In this context, while Sph2 from *L. interrogans* serovar Bataviae; *L. alstonii* serovars Pingchang and Sichuan; *L. borgpetersenii* serovars Pomona and Hardjo-bovis; *L. noguchii* serovars Panama and Autumnalis; and *L. santarosai* serovar Arenal presented no modifications in amino acid residues in the catalytic or metal-binding sites, *L. interrogans* serovar Australis presented six (67%) different residues. Additionally, there were eight surface-exposed amino acids described as being associated with the interaction with the host cell membrane (W172, Y242, W274, F275, Y382, Y383, Y384, and Y425). Interestingly, these amino acids seemed to be less conserved among pathogenic leptospires, only one out of the three known serovars (*L. interrogans* serovar Bataviae, and *L. noguchii* serovars Panama and Autumnalis) presented no changed amino acids in the alignment, while the other studied proteins presented from three to six changed amino acids. Additionally, when compared with sphingomyelinases from other bacteria, Sph2 presented differences only in amino acids involved with the interaction with host cells, while all amino acids of the catalytic site and metal-binding sites remained unaltered (Table 4).

Regarding the location of Sph2 epitopes, only the epitope Sph2_(176-191)_ was inserted in Sphingomyelinase C domain (Sph2_(155-440)_). Finally, aiming to allow the visualization of predicted epitopes, catalytic and metal-binding sites, and amino acids associated with the interaction with the host cell membrane in the Sph2 3D structure, we highlight these structures in the Alphafold predicted model (ID: AF-P59116-F1) (Figure 1).

### 3.5. Studied Population

The studied population was composed of 87 Brazilian febrile suspected leptospirosis cases based on their contact with rodents, floods, or other risk factors. Among them, 51 patients were reactive to *Leptospira* spp. (MAT positive), presenting antibody titers on MAT ranging from 1:800 to 1:12800 (mean: 1:2839 ± 2407), and were grouped as reactive against *Leptospira* spp. (RL), while 36 patients were non-reactive against *Leptospira* spp. (NRL).

The studied population presented was 35.7 (±17.5) mean years old, and clinical and epidemiological data were obtained between 1 and 74 days after the beginning of symptoms (mean 12.5 ± 12). Our studied group was mostly composed of men (85%), but we did not observe statistical differences in the frequency of men in RL (90%) and NRL (78%) (*p* = 0.134). Moreover, in both groups, RL and NRL, about 11.5% of deaths cases were reported. As shown in Table 5, these groups were also similar in age and days of symptoms.

Regarding reported symptoms, fever (84%), myalgia (77%), and headache (63%) were the most prevalent symptoms among studied patients. However, comparing the frequencies of symptoms among groups, only jaundice (RL = 68.6% and NRL = 41.7%; *p* = 0.0159), calf pain (RL = 60.8% and NRL = 41.7%, *p* = 0.0257), renal insufficiency (RL = 39.2% and NRL = 13.9%, *p* = 0.0463), and pulmonary hemorrhage (RL = 13.7% and NRL = 0%, *p* = 0.0382) were statistically more frequent in RL patients.

In the RL group, only one *Leptospira* serovar was detected by MAT using serum samples of 35 patients (69%), while the samples of 14 individuals (27%) cross-reacted with two serovars, one individual sample (2%) cross-reacted with serovars Australis, Hebdomadis, and Autumnalis, and another (2%) recognized the serovars Copenhageni, Canicola, Icterohaemorrhagiae, and Tarassovi (Table 5). Regarding diagnosed serovars in studied patients, Tarassovi and Copenhageni were the most prevalent serovars, each of them diagnosed in about 47% of studied patients, singly reported in 31.4% and 23.5%, respectively, and in 15.7% and 23.5% of patients whose sera recognized two or more serovars.

### 3.6. Naturally Acquired Antibodies against Sph2(176-191) and Sph2(446-459)

We assessed the naturally acquired IgG and IgM response against the synthetic peptides containing the sequences of epitopes Sph2_(176-191)_ and Sph2_(446-459)_ in plasma samples from 51 confirmed leptospirosis patients (RL: MAT reactive to *Leptospira* spp.) and 36 non-confirmed cases (NRL: non-reactive to *Leptospira* spp.). First, using the defined cut-off, the predicted epitopes were specifically recognized by serum samples of the patients from the RL group, as the NRL samples were non-responders against the predicted epitopes (Figure 2a). The epitopes Sph2_(176-191)_ and Sph2_(446-459)_ were specifically recognized by 33.3% (n = 17) and 19.6% of RL individuals, respectively, with a prevalence of IgG immune response to both epitopes. As shown in Figure 2b, 17 (33.3%) and 9 (17.6%) individuals presented IgG antibodies against Sph2_(176-191)_ and Sph2_(446-459)_, while only one (2%) and two (3.9%) patients presented IgM antibodies, respectively (Figure 2b). Among responders against epitopes, only one individual exclusively presented IgM antibodies against Sph2_(446-459)_, while the other two IgM responders also presented IgG antibodies. Moreover, among 20 individuals that presented antibodies against at least one Sph2 epitope, ten individuals only presented antibodies against Sph2_(176-191)_, three against Sph2_(446-459)_, and seven individuals presented antibodies against both epitopes. Regarding the magnitude response, there were no statistical differences between IgG reactivity indexes against each peptide, which ranged from 1.05 to 3.23 (median: 1.27) against Sph2_(176-191)_ and ranged from 1.07 to 1.81 (median: 1.19) against Sph2_(446-459)_ (Figure 2c).

Regarding the detection of antibodies in patients infected by different serovars, we detected specific antibodies against Sph2_(176-191)_ in patients reactive to serovars: Tarassovi (*n* = 6), Copenhageni (*n* = 3), Canicola (*n* = 1), Copenhageni/ Icterohaemorrhagiae (*n* = 3), Copenhageni/Tarassovi (*n* = 2), Tarassovi/Icterohaemorrhagiae (*n* = 1), and Copenhageni/Hebdomadis (*n* = 1). Furthermore, specific antibodies against Sph2_(446-459)_ were detected in patients reactive to serovars: Tarassovi (*n* = 4), Copenhageni (*n* = 2), Grippotyphosa (*n* = 1), Copenhageni/Icterohaemorrhagiae (*n* = 2), and Copenhageni/Tarassovi (*n* = 1).

### 3.7. Associations between Humoral Response and Clinical Features

Based on the observed differences in the frequencies of symptoms between RL and NRL patients, we further explored the frequencies of cases presenting jaundice, renal insufficiency, pulmonary hemorrhage, calf pain, and deaths by comparing the frequencies of these symptoms between individuals whose sera recognized or not the epitopes Sph2_(176-191)_ and Sph2_(446-459)_. In this context, considering that pulmonary hemorrhage was a symptom exclusively observed among RL patients, we highlight that its frequency was statistically higher among responders to Sph2_(176-191)_ (29.4%) than among non-responders (5.9%; *p* = 0.0338). However, this difference was not observed among responders (20%) and non-responders (12.2%) to Sph2_(446-459)_ (*p* = 0.6116).

As shown in Figure 3, despite seeming to be higher, the frequencies of death cases among responders to Sph2_(176-191)_ (23.5%) and Sph2_(446-459)_ (30%) were not statistically different when compared with non-responders to Sph2_(176-191)_ (5,9%; *p* = 0.0865) and Sph2_(446-459)_ (7.3%, *p* = 0.0812), and neither when compared with the observed frequency of death cases in the NRL group (11.1%, *p* = 0.2518 and *p* = 0.1632, respectively). Regarding other symptoms, we did not observe statistical differences between responders and non-responders to synthetic peptides. Remarkably, the frequency of renal insufficiency cases among non-responders to Sph2_(446-459)_ (41.5%) was statistically higher than the frequency observed in NRL (13.9%, *p* = 0.109), while the frequency among responders to Sph2_(446-459)_ (30%) was statistically similar to both non-responders (*p* = 0.7208) and NRL (*p* = 0.3442). Additionally, responders and non-responders to Sph2_(176-191)_ presented similar frequencies of renal insufficiency (41.2% and 38.2%, *p* > 0.9999), both higher than the NRL group (*p* = 0.0382 and *p* = 0.0285, respectively). Additionally, we observed higher frequencies of jaundices among responders to Sph2_(176-191)_ (82.4%, *p* = 0.0076) and non-responders to Sph2_(446-459)_ (70.7%, *p* = 0.0123) than among NRL (11.1%), and observed higher frequencies of calf pain among responders to Sph2_(176-191)_ (76.5%) than NRL (41.7%, *p* = 0.0217). Remarkably, clinical complications, such as pulmonary hemorrhage, were only observed in individuals with low or no-reactivity against Sph2_(176-191)_, while individuals with high reactivity indexes to this epitope presented only mild symptoms.

## 4. Discussion

Sphingomyelin (SM) is a major constituent of eukaryotic cell membranes and the ability to degrade this phospholipid by bacteria may consequently contribute to the pathogenesis of infection. Sphingomyelinases are a group of hemolysins, present in both eukaryotes and prokaryotes, which are related to phospholipid metabolism in the former and that frequently act as toxins in the latter [43], that are absent in non-pathogenic *Leptospira* spp. [17,44]. In pathogenic *Leptospira*, sphingomyelinases play an important role in their survival in the mammalian host by mediating the lysis and release of essential nutrients from the host cells [18], but are implicated in the hemorrhagic complications associated with leptospirosis [45,46]. Among sphingomyelinases described in *L. interrogans* serovar Lai, Sph2 and SphH are proven to have cytotoxic properties and are expressed during infection [19,47]. However, studies focused on the associations of sphingomyelinases with leptospirosis severity, their potential as protective antigens, and the identification of their B-cell epitopes, remain scarce. To the best of our knowledge, this was the first study aiming to identify linear B-cell epitopes in *L. interrogans* serovar Lai Sph2 and to explore their association with the clinical data of naturally infected patients.

First, we used a combination of algorithms to predict linear B-cell epitopes in *L. interrogans* serovar Lai Sph2. This approach had been used and improved by our group in recent years to predict epitopes in viruses [25,26], bacteria [24], and protozoans [28,48,49], resulting in prediction accuracy up to 90% in the most recent studies. Here, we predicted two sequences as antigenic and linear B-cell epitopes: GHDERAKRISKSDYVK (Sph2_(176-191)_) and TPTKSGHKKKYDQV (Sph2_(446-459)_), which did not present similar epitopes described in BLASTP, in the Immune Epitope Database http://www.iedb.org/home_v3.php (accessed on 20 June 2020) (data not shown), and also did not present similar sequences described in humans and mice, or in other databases from PeptideAtlas http://www.peptideatlas.org/map/ (accessed on 20 June 2020). Therefore, predicted epitopes were considered non-conserved among hosts and other bacteria, suggesting them as antibody targets specific against *Leptospira* spp. In this context, when compared with other sphingomyelinases from *L. interrogans* serovar Lai, we observed that Sph2 is highly similar to Sph4, presenting more than 99% of identity, while other sphingomyelinases presented identities from 53% to 66.4% (Table 3). Moreover, Sph2 is highly variable among pathogenic leptospires, with identities that ranged from 55% to 90%. Despite the similarity observed between *L. interrogans* serovar Lai and serovar Bataviae (identity of 90%), Sph2 from serovar Lai was more similar to *L. noguchii* serovars Panama and Autumnalis (identities of 89.6%) than to other serovars of *L. interrogans* (Lora, Zanoni, and Pyrogenes), corroborating studies that recognized the great numbers of serovars as an obstacle that has hampered the development of a universal vaccine for leptospirosis [50].

Based on these data and aiming for the future constructions of multi-epitope vaccines, we explored the conservation of predicted epitopes among other sphingomyelinases from *L. interrogans* serovar Lai and Sph2 from other pathogenic leptospires. Remarkably, the epitope Sph2_(446-459)_ was highly conserved among Sph2 from pathogenic leptospires and among sphingomyelinase from *L. interrogans* serovar Lai, presenting more than 71.4% of identity when compared with studied sequences. However, Sph2_(176-191)_ was highly conserved in only three studied Sph2 (*L. interrogans* serovar Bataviae and *L. noguchii* serovars Panama and Autumnalis) and in Sph4, presenting 100% of identity with these proteins, while presenting less than 56.3% of identity when compared with other studied proteins. From our point of view, the high conservation of Sph2_(446-459)_ among pathogenic leptospires suggests that this epitope may be inserted in a region under low selective pressure by the host immune response. In line with this assumption, according to the prediction of algorithms InterPro family, TIGRFAMs, and nSMase (data not shown), this epitope is located out of the Sph2 Sphingomyelinase C domain (Sph2_(155-440)_), supporting the hypothesis of low selective pressure given that all amino acids involved in Sph2 activities are located on the Sphingomyelinase C domain [18], a common exo-endo-phosphatase domain that classifies sphingomyelinases in the DNase I superfamily, which differs in structure and substrate specificity [51].

In the same way, we evaluated the variability of amino acids in the catalytic site, metal-binding site, and the region of interaction with host membranes (Table 5). First, we highlighted that Sph2 from *L. interrogans* serovar Lai presents all amino acids described in the catalytic and metal-binding sites highly conserved (100% of identity) when compared with sphingomyelinases from other bacteria, such as *Listeria ivanovii* and *Bacillus cerus*, which are considered membrane-damaging virulence factors that induce hemolysis, a reduction in phagocytosis [52], and escape from the phagocytic vacuole [53]. Additionally, our data showed high variability in amino acids associated with the host membrane interaction when compared with other bacterial sphingomyelinases and when compared with Sph2 from other leptospires. These data suggest that Sph2 from different serovars may play a role in host tropism and the pathogenicity in different hosts, corroborating the hypothesis of Gonzáles-Zorn and collaborators, to scmL [53], which is structurally close to Sph2 from *L. interrogans* serovar Lai [18], and propose an additional explanation for the inability of Sph2 from *L. interrogans* serovar Copenhageni Fiocruz L1-130 to lyse sheep erythrocytes [20]. Based on these data, we explored the location of predicted epitopes in the Sph2 3D structure (Figure 3), observing that epitope Sph2_(176-191)_ was closely located to the catalytic site, metal-binding site, and the region related to interaction with the host membrane. This finding suggests that antibodies against Sph2_(176-191)_ could hamper the functionality of Sph2 by blocking the interaction with the host, or even the ligation to cofactor or substrate. However, studies aiming to identify the active sites of Sph2 from *Leptospira* spp. and to evaluate their activity and the role of antibodies against their epitopes remain a lack in the literature.

Until now, antibodies against Sph2 were detected in the blood of mares following leptospiral abortion, but not in horses immunized with bacterins [21], and in serum from leptospirosis convalescent patients [20]. These studies confirmed the protein expression during leptospirosis but lacked the investigation of associations between the presence of antibodies and clinical data. To the best of our knowledge, this was the first study aiming to identify Sph2 B-cell epitopes and investigate their association with clinical data. Here, we evaluated the presence of antibodies against epitopes Sph2_(176-191)_ and Sph2_(446-459)_ in blood samples from 51 Brazilian patients with leptospirosis, which presented reactivity (or cross-reactivity) to 14 *Leptospira* spp. serovars, with the prevalence of Tarassovi and Copenhageni. Interestingly, this observation is in disagreement with the recent revision of Browne and collaborators, which reported the serovars Icterohaemorrhagiae, Canicola, and Pomona as the most prevalent in human leptospirosis in the Americas from 1930 to 2017, with Pomona and Canicola being the most prevalent in Brazil [54]. From our point of view, this controversy reinforces the necessity of improved surveillance of leptospirosis cases, aiming to know what are the most important pathogens to focus on in vaccine development.

In this context, thinking about reactivity and cross-reactivity to predicted antibodies, first, we confirmed that the predicted epitopes were naturally immunogenic, since about 40% of serum samples from patients naturally infected by *Leptospira* spp. were able to recognize at least one of these epitopes, with a prevalence of IgG antibodies among responders to both Sph2_(176-191)_ and Sph2_(446-459)_. Both epitopes Sph2_(176-191)_ and Sph2_(446-459)_ were recognized by patients’ samples reactive to at least one of the most prevalent serovars (Tarassovi, Copenhageni, and Icterohaemorrhagiae), while Sph2_(176-191)_ was also recognized by patients’ samples reactive to serovars Canicola and Hebdomadis, and Sph2_(446-459)_ was also detected in samples reactive to serovar Grippotyphosa. Remarkably, among the serovars detected in our studied population, only samples reactive to serovars Wolffi (n = 1), Pomona (n = 1), Wolffi/Sejroe (n = 1), and Australis/Hebdomadis/Autumnalis (n = 1) presented no reactivity to at least one of predicted epitopes. However, considering the low level of identity shared by *L. interrogans* serovar Lai Sph2_(176-191)_ and other pathogenic species, we cannot discard a higher frequency of responder and reactivity index investigating the epitopes from other important serovars. Therefore, the real cross-reactivity of these epitopes needs to be explored more in serological and epidemiological studies, since the absence of reactivity could be related to the limited number of studied patients and the expression of Sph2, which is markedly upregulated by changes in osmolarity and temperature [55], determining different levels of expression and clinical outcomes in distinct hosts and patients.

Moreover, the protective potential of antibodies against Sph2 is still unknown. Regarding its potential as a vaccine antigen, hamsters immunized with the recombinant Sph2 from *L. interrogans* serovar Copenhageni did not present protection when challenged with a virulent strain of *L. interrogans* serovar Pomona; however, the authors propose that this can be related to a lack of correct folding of the recombinant proteins [20]. Unfortunately, due to the limited number of patients involved in this study, we believe that our data are not sufficient to confirm or discard the protective role of antibodies against Sph2 epitopes. Here, we observed a higher frequency of cases of pulmonary hemorrhage among responders to Sph2_(176-191)_ (Figure 2). We believe that this finding could be indicative of the association of Sph2 with hemorrhages in leptospirosis since only individuals with low reactivity (R.I. < 2) to Sph2 epitopes presented clinical complications (pulmonary hemorrhage, renal insufficiency, or death), while high responders to Sph2_(176-191)_ presented only mild symptoms. Based on this data and on the position of Sph2_(176-191)_ close to the catalytic site and the region of interaction of Sph2 with the host membrane, we conjecture that this epitope should be better explored as a target of protective antibodies against leptospirosis; however, more studies are necessary to prove this hypothesis. On the other hand, we also hypothesized that different levels of Sph2 expression related to *Leptospira* species and host factors can also be related to reactivity against Sph2 epitopes and could be associated with leptospirosis hemorrhagic symptoms, reinforcing the necessity of more studies to prove Sph2 and its epitopes as vaccine candidates.

Up to now, the main vaccines proposed against *Leptospira* elicit a serovar-dependent immunity [50]. Though, considering that there are more than 300 classified serovars, the development of a universal vaccine for leptospirosis persists as a great challenge [10]. Moreover, inactivated vaccines do not promote long-term protection, and some side effects have been reported [56], supporting the necessity of novel strategies for vaccine development, such as multi-epitope vaccines. This approach is based on the rational combination of epitopes from proteins previously classified as vaccine candidates and has been used in an increasing number of studies aiming to propose protective vaccines for leptospirosis [57,58,59,60,61]. However, the number of epitopes from *Leptospira* antigens experimentally validated remains scarce, and justifies our study, since the identification of protective epitopes is the key to the design of an effective and universal multi-epitope vaccine for human leptospirosis.

In brief, we used immunoinformatics to predict linear B-cell epitopes in Sph2 from *L. interrogans* serovar Lai and to assess their natural immunogenicity in human leptospirosis. However, further studies using patients from different regions and additional *Leptospira* spp. serovars are necessary to investigate the real protective role of antibodies against epitopes Sph2_(176-191)_ and Sph2_(446-459)_ in leptospirosis_._ Moreover, studies in animal models are essential to prove their potential as a vaccine antigen, singly and combined with other identified epitopes.

## 5. Conclusions

Our study was the first to identify antibody targets in Sphingomyelinase 2 from *L. interrogans* serovar Lai. Our data corroborate the association of Sph2 with hemorrhagic complications from leptospirosis. Moreover, based on its location and associations with clinical data and specific immune response, we suggest that Sph2_(176-191)_ can be further explored in multi-epitope vaccines to prove its protective potential in animal models.

## Figures and Tables

**Figure 1 vaccines-11-00359-f001:**
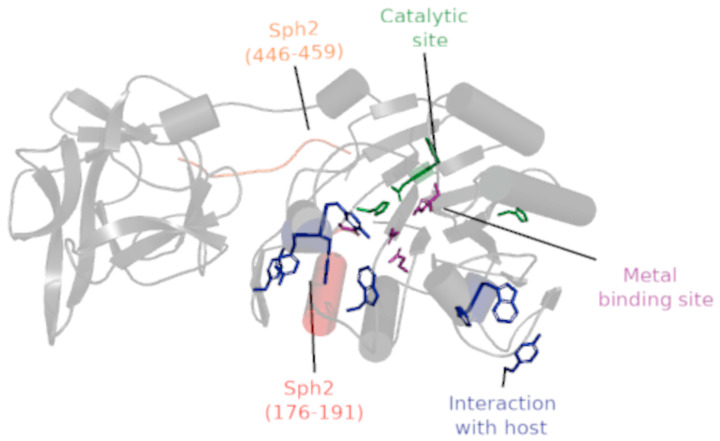
*L. interrogans* serovar Lai Sph2 3D-structure (AF-P59116-F1). The protein chain is indicated by a gray cartoon, where the catalytic site, metal-binding site, and amino acids associated with the interaction with the host are represented as licorice in green, purple, and blue, respectively. The locations of epitopes Sph2_(176-191)_ and Sph2_(446-459)_ are highlighted in red and orange, respectively. In the cartoon, cylindrical helices, round helices, flat sheets, and smooth loops are applied to allow better visualization of Alpha-Fold predicted protein AF-P59116-F1_(155-612)_.

**Figure 2 vaccines-11-00359-f002:**
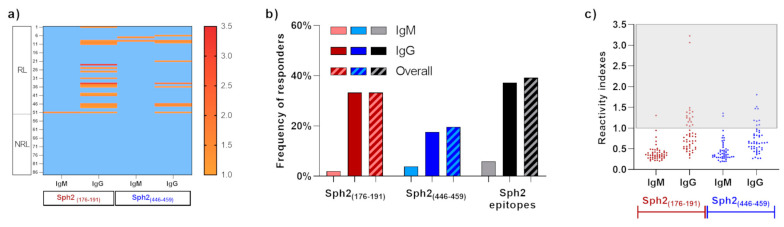
Evaluation of natural immunogenicity of predicted epitopes. (**a**) Heatmap of IgM and IgG reactivity indexes against synthetic epitopes. Values higher than 1 represent responder individuals and were indicated in the color scale, and non-responders were indicated by a light blue color. (**b**) Frequencies of IgM (light color), IgG (dark color), and overall (striped bar) responders to Sph2_(176-191)_ (red bars), Sph2_(446-459)_ (blue bars), and responders to at least one Sph2 epitope (gray/black bars). (**c**) IgM and IgG reactivity indexes against Sph2_(176-191)_ (red points) and Sph2_(446-459)_ (blue points). Responders to peptides were highlighted by the gray square.

**Figure 3 vaccines-11-00359-f003:**
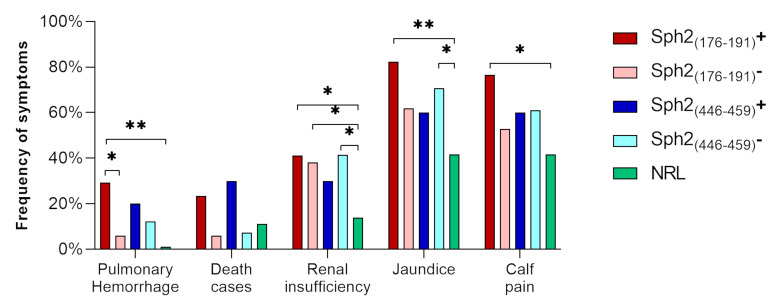
Frequency of symptoms and complications observed in responders (dark colors) and non-responders (light colors) to Sph2_(176-191)_ (red bars), Sph2_(446-459)_ (blue bars), and NRL (green bars). The frequency of symptoms was defined by the ratio between the number of patients presenting the symptom and the total number of patients in the group. Statistical differences, evaluated by the Chi-square test, are indicated by asterisks: (*): *p* < 0.05; (**): *p* < 0.001.

**Table 1 vaccines-11-00359-t001:** General data of studied proteins.

Protein	Species	Serogroup	Serovar	Uniprot ID	Length (a.a.)	Mass (Da)
Sph2	*L. interrogans*	Icterohaemorrhagiae	Lai	P59116	623	71030
		Australis	Lora	M3E462	507	58766
		Bataviae	Bataviae	M6TDN9	423	48137
		Pyrogenes	Zanoni	M6I333	507	58766
		Pyrogenes	Pyrogenes	M7AA65	567	64539
	*L. alexanderi*	Manhao	Manhao 3	V6I1S1	458	52886
	*L. alstonii*	Ranarum	Pingchang	T0FYI7	543	61077
		Undesignated	Sichuan	M6D0Q2	543	61077
	*L. borgpetersenii*	Pomona	Pomona	M6W693	566	64409
		Sejroe	Hardjo-bovis	M6BNC3	556	63269
	*L. noguchii*	Panama	Panama	T0FTK3	616	70064
		Autumnalis	Autumnalis	M6UEI2	591	67772
	*L. kirschneri*	Autumnalis	Bulgarica	M6F3J0	559	64827
	*L. santarosai*	Javanica	Arenal	M6JW85	622	70525
	*L. weilii*	Ranarum	Ranarum	N1WDQ5	664	74211
		Tarassovi	Topaz	M3FRC5	758	81501
SphH	*L. interrogans*	Icterohaemorrhagiae	Lai	O34095	554	64433
SpH1	*L. interrogans*	Icterohaemorrhagiae	Lai	P59115	597	68192
Sph3	*L. interrogans*	Icterohaemorrhagiae	Lai	A0A0E2DC81	596	68095
Sph4	*L. interrogans*	Icterohaemorrhagiae	Lai	A0A0E2DCF7	623	70995
smcL	*L. ivanovii*			Q9RLV9	335	38455
BCSmase	*B. cereus*			P09599	333	36949
Sph	*S. aureus*			A0A7U4AUV1	330	37238
SphH	*Pseudomonas sp.*			Q93HR5	516	58114

**Table 2 vaccines-11-00359-t002:** Amino acids predicted by each algorithm in Sph2 epitopes.

	Sph2_(176-191)_	Sph2_(446-459)_
Predicted Sequence	GHDERAKRISKSDYVK	TPTKSGHKKKYDQV
Bepipred	GHDERAKRISKSDYVK	TPTKSGHKKKYDQV
ESA	GHDERAKRISKSDYVK	TPTKSGHKKKYDQV
ABCpred	GHDERAKRISKSDYVK	TPTKSGHKKKYDQV
ElliPro	GHDERAKRISKSDYVK	TPTKSGHKKKYDQV
BCePred-ERS	GHDERAKRISKSDYVK	TPTKSGHKKKYDQV
BCePRed-HS	GHDERAKRISKSDYVK	TPTKSGHKKKYDQV
BCePRed-PS	GHDERAKRISKSDYVK	TPTKSGHKKKYDQV
Lbtope	GHDERAKRISKSDYVK	TPTKSGHKKKYDQV
COBEpro	GHDERAKRISKSDYVK	TPTKSGHKKKYDQV

Black letters represent amino acids predicted by the algorithm and gray letters indicate non-predicted amino acids by each used algorithm.

**Table 3 vaccines-11-00359-t003:** Comparison of Sph2 and predicted epitopes among leptospires and other sphingomyelinases.

Gene	Specie	Serogroup	Serovar	Sph2	Sph2_(176-191)_		Sph2_(446-459)_	
				Identity	Identity	GHDERAKRISKSDYVK	Identity	TPTKSGHKKKYDQV
Sph2	*L. interrogans*	Australis	Lora	55.0%	43.8%	.QEK..RLLVD.K...	71.4%	........R....I
		Bataviae	Bataviae	90.0%	100.0%	................	85.7%	...Q.A.RR.....
		Pyrogenes	Zanoni	55.0%	43.8%	.QEK..RLLVD.K...	71.4%	...Q.A.RR.....
		Pyrogenes	Pyrogenes	67.1%	56.3%	.QE...Q..AS.S.I.	92.9%	........R.....
	*L. alexanderi*	Manhao	Manhao 3	73.2%	50.0%	.QN...Q..VS.N.IQ	78.6%	......R.R...R.
	*L. alstonii*	Ranarum	Pingchang	69.7%	43.8%	AQN...EL.AS..HI.	85.7%	.......RR.....
		Undesignated	Sichuan	69.7%	43.8%	AQN...EL.AS..HI.	85.7%	.......RR.....
	*L. borgpetersenii*	Pomona	Pomona	70.6%	50.0%	.QN...Q..VS.N.IQ	78.6%	......R.R...R.
		Sejroe	Hardjo-bovis	70.6%	50.0%	.QN...Q..VS.N.IQ	78.6%	......R.R...R.
	*L. noguchii*	Panama	Panama	89.6%	100.0%	................	85.7%	...E....R.....
		Autumnalis	Autumnalis	89.6%	100.0%	................	85.7%	........R....I
	*L. kirschneri*	Autumnalis	Bulgarica	47.2%	31.3%	.QE...NLLLN.QHIQ.	78.6%	...F.T......R.
	*L. santarosai*	Javanica	Arenal	69.3%	50.0%	.QN...E..AS.N.IR	85.7%	........R...R.
	*L. weilii*	Ranarum	Ranarum	64.1%	43.8%	.QND..E..ASAN.I.	92.9%	........R.....
		Tarassovi	Topaz	57.6%	56.3%	.QK...EQ.AN...I.	85.7%	.......RR.....
SphH	*L. interrogans*	Icterohaemorrhagiae	Lai	53.0%	43.8%	.QEK..RLLVD.K...	71.4%	...Q.A.RR.....
Sph1	*L. interrogans*	Icterohaemorrhagiae	Lai	66.4%	56.3%	.QE...Q..AS.S.I.	92.9%	........R.....
Sph3	*L. interrogans*	Icterohaemorrhagiae	Lai	66.4%	56.3%	.QE...Q..AS.S.I.	92.9%	........R.....
Sph4	*L. interrogans*	Icterohaemorrhagiae	Lai	99.3%	100.0%	................	100.0%	..............
smcL	*L. ivanovii*	-	-	46.0%	43.8%	.QMH..DL.AQA..M.	28.6%	..VIGRS.HGW.KT.
Sph	*B. cereus*	-	-	49.0%	43.8%	.QSQ..DL.GAA..I.	35.7%	.V.SWLK.YT..DY
Sph	*S. aureus*	-	-	48.7%	43.8%	.QYK..DL.GQ.S.I.	35.7%	KK...NSL..VATL
SphH	*Pseudomonas sp.*	-	-	42.0%	37.5%	AL.IPSP.W.A..G.N	28.6%	DNDQGQCL.DG...

**Table 4 vaccines-11-00359-t004:** Amino acids in pathogenic leptospires Sph2 catalytic sites, metal-binding sites, and amino acids involved in host membrane interactions.

*Specie* (Serovar)-Protein	Catalytic Site				Amino Acids in the Central Metal-Binding Site					Surface-Exposed Amino Acids Involved with the Host Membrane Interaction							
*L. interrogans* (Lai)-Sph2	H293	D393	Y394	H433	N161	E200	D341	N343	D432	W172	Y242	W274	F275	Y382	Y383	Y384	Y425
*L. interrogans* (Australis)-Sph2	H230	T328	W398	-	S98	G137	D278	N280	-	W109	S179	Q211	Y212	Y319	Y320	Y321	Y337
*L. interrogans* (Bataviae)-Sph2	H236	D336	Y337	H376	N104	E143	D284	N286	D375	W115	Y185	W217	F218	Y325	Y326	Y327	Y368
*L. interrogans* (Lora)-Sph2	H175	E274	Y274	R314	N40	G79	S221	N223	D313	S51	F121	Y156	Y157	Y262	T263	S264	Y306
*L. interrogans* (Zanoni)-Sph2	H175	E274	Y274	R314	N40	G79	S221	N223	D313	S51	F121	Y156	Y157	Y262	T263	S264	Y306
*L. interrogans* (Pyrogenes)-Sph2	H232	D332	Y332	R373	S100	G139	D280	N282	D372	W111	S181	Q213	Y214	Y321	Y322	Y323	Y365
*L. alexanderi* (Manhao3)-Sph2	H126	D226	Y226	H266	-	E33	D174	N176	D265	W5	D75	F107	F108	F215	Y216	Y217	Y258
*L. alstonii* (Pingchang)-Sph2	H211	D311	Y311	H351	N79	E118	D259	N261	D350	W90	L160	Y192	F193	F300	Y301	Y302	Y343
*L. alstonii* (Sichuan)-Sph2	H211	D311	Y311	H351	N79	E118	D259	N261	D350	W90	L160	192Y	F193	F300	Y301	Y302	Y343
*L. borgpetersenii* (Pomona)-Sph2	H234	D334	Y334	H374	N102	E141	D282	N284	D373	W113	D183	V215	F216	F325	Y326	Y327	Y366
*L. borgpetersenii* (Hardjo-bovis)-Sph2	H224	D324	Y324	H364	N92	E131	D272	N274	D363	W103	D173	V205	F206	F313	Y314	Y315	Y356
*L. noguchii* (Panama)-Sph2	H286	D386	Y386	H426	N154	E193	D334	N336	D425	W165	Y235	W267	F268	Y375	Y376	Y377	Y418
*L. noguchii* (Autumnalis)-Sph2	H261	D361	Y361	H401	N129	E168	D309	N311	D400	W140	Y210	W242	F243	Y350	Y351	Y352	Y393
*L. kirschneri* (Bulgarica)-Sph2	H224	N329	Y330	Y369	N88	G127	N274	G276	D368	T99	T170	Y203	F204	Y315	L316	Q317	Y361
*L. santarosai* (Arenal)-Sph2	H290	D390	Y390	H430	N158	E197	D338	N340	D429	W169	E239	F271	F272	F379	Y380	Y381	Y422
*L. weilii* (Ranarum)-Sph2	H331	D431	Y431	Y472	N199	R238	N379	N381	D471	W210	S280	Q312	Y313	F420	K421	Y422	Y464
*L. weilii* (Topaz)-Sph2	H421	N521	Y522	S562	N289	G328	D469	N471	D561	W300	S370	S402	S403	L510	K511	Y512	H554
*L. interrogans* (Lai)-SphH	H222	E321	Y322	R361	N87	G126	S268	N270	D360	S98	F168	Y203	Y204	Y309	T310	S311	Y353
*L. interrogans*(Lai)-Sph1	H262	D362	Y363	R403	S130	G169	D310	N312	D402	W141	S211	Q243	Y244	Y351	Y352	Y353	Y395
*L. interrogans* (Lai)-Sph4	H261	D361	Y362	R402	S129	G168	D309	N311	D401	W140	S210	Q242	Y243	Y350	Y351	Y352	Y394
*L. interrogans* (Lai)-Sph4	H293	D393	Y394	H433	N161	E200	D341	N343	D432	W172	Y242	W274	F275	Y382	Y383	Y384	Y425
*L. ivanovii*-smcL	H180	D282	Y283	H325	N51	E88	D229	N231	D324	-	F130	R161	L162	E271	S272	Y273	Y317
*B. cereus*-Sph	H173	D280	Y281	H323	N43	E80	D222	N224	D322	-	S123	N154	L155	Y269	N270	F271	Y315
*S. aureus*-Sph	H178	D279	Y280	H322	N49	E86	D227	N229	D321	-	T128	N159	D160	Y268	N269	Y270	Y314
*Pseudomonas sp.*-SphH	H166	D277	Y278	H320	N37	E75	D214	N216	D319	F48	N117	R147	L148	Y256	Q257	Y258	W312

Non-conserved amino acids with *L. interrogans* serovar Lai are indicated by gray cells.

**Table 5 vaccines-11-00359-t005:** Clinical and epidemiological features of the study population.

Epidemiological and Clinical Data	RL (n = 51)	NRL (n = 36)	
	n (%)		
Male	46 (90.2%)	28 (77.8%)	
Female	5 (9.8%)	8 (22.2%)	
Deaths	6 (11.8%)	4 (11.1%)	
	Median (Interquartile Range)		
Age	37 (22–49)	36.5 (17–45)	
Days of Symptoms	9 (6–15)	7 (5–16)	
Symptoms	n (%)		*p*-value
Fever	44 (86.3%)	29 (80.6%)	
Myalgia	43 (84.3%)	24 (66.7%)	
Headache	34 (66.7%)	21 (58.3%)	
Jaundice	35 (68.6%)	15 (41.7%)	0.0159
Calf Pain	31 (60.8%)	15 (41.7%)	0.0257
Renal Insufficiency	20 (39.2%)	5 (13.9%)	0.0463
Pulmonary Hemorrhage	7 (13.7%)	0 (0%)	0.0382
Hemorrhagic Signs	8 (15.7%)	2 (5.6%)	
Prostration	27 (52.9%)	18 (50.0%)	
Vomit	24 (47.1%)	18 (50.0%)	
Diarrhea	20 (39.2%)	11 (30.6%)	
Respiratory Changes	18 (35.3%)	10 (27.8%)	
Conjunctival Congestion	10 (19.6%)	8 (22.2%)	
Abdominal Pain	4 (7.8%)	5 (13.9%)	
Diagnosed *Leptospira* spp. Serovars	n (%)		
Tarassovi	16 (31.4%)		
Copenhageni	12 (23.5%)		
Grippotyphosa	2 (3.9%)		
Canicola	2 (3.9%)		
Icterohaemorrhagiae	1 (2%)		
Wolffi	1 (2%)		
Pomona	1 (2%)		
Tarassovi/Copenhageni	5 (9.8%)		
Copenhageni/Icterohaemorrhagiae	5 (9.8%)		
Tarassovi/Icterohaemorrhagiae	1 (2%)		
Copenhageni/Hebdomadis	1 (2%)		
Tarassovi/Grippotyphosa	1 (2%)		
Wolffi/Sejroe	1 (2%)		
Australis/Hebdomadis/Autumnalis	1 (2%)		
Copenhageni/Canicola/Icterohaemorrhagiae/Tarassovi	1 (2%)		

RL: reactive to *Leptospira* spp. (MAT positive), NRL: non-reactive to *Leptospira* spp. (MAT negative).

## Data Availability

Data are available on request due to restrictions of privacy or ethical. The data presented in this study are available on request from the corresponding author. The data are not publicly available due to confidential information related to the personal information of donors, in accordance with the Institutional Ethics Committee of the Oswaldo Cruz Foundation.

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
