# Peer review of "Sph2_(176–191)_ and Sph2_(446–459)_: Identification of B-Cell Linear Epitopes in Sphingomyelinase 2 (Sph2), Naturally Recognized by Patients Infected by Pathogenic Leptospires"

_vaccines, 2023, doi:10.3390/vaccines11020359_

Round 1
Reviewer 1 Report
This research paper focuses the ‘Sph2(176-191) and Sph2(446-459): identification of B-cell linear 2 epitopes in Sphingomyelinase 2 (Sph2), naturally recognized 3 by patients infected by pathogenic leptospires’ . In my opinion, it is a good research paper. They have done a lot of works. The data are presented and explained very well.
Author Response
Dear Reviewer,
Firstly, we want to thank you for your time and efforts in helping us to improve the manuscript quality. We appreciated your comment and try to improve the article with all the received suggestions.
Best regards,
Reviewer 2 Report
The author identified two B-cell epitopes, Sph2(176-191) and Sph2(446-459), at sphingomyelinase 2 (sph2) of Leptospira spp. by immunoinformatics approaches. Furthermore, the authors tested the antibody reactivity to the two B-cell epitope peptides of the sera from leptospirosis patients. The authors declared that high responders to Sph2(176-191) had only mild symptoms in the clinical investigation, indicating a protective role of the humoral immune response to Sph2(176-191). The authors found a candidate protective B-cell epitope and their study is helpful to the design of leptospirosis vaccines. Here are some suggestions and questions to improve their article.
1) Would the authors add some information on the current development of leptospirosis vaccines, in the part of introduction? It would be helpful for the readers to understand the background and significance of their study.
2) Would the authors provide more discussion on the leptospirosis patients who did not respond to Sph2(176-191) nor Sph2(446-459)? Did these patient respond to recombinant Sph2 protein? Considering the amino acid sequences of Sph2(176-191) in some Leptospira spp. are not conservative (Table 3), the authors should discuss the influence of the sequence of Sph2(176-191) to the serum responses in the patient population.
3) The authors declared that clinical information implied the protective potential of Sph2(176-191) epitope. However, I can not read it from Fig. 3. The statistics on frequency of symptoms are quite controversial. The authors should revise the chapter 3.7 Associations between Humoral Response and Clinical Features to make it support their conclusion.
Moreover, I suggested the authors provide some direct evidence about the protective ability of the antibodies specific for Sph2(176-191) if possible. For example, they can test the antibodies induced in mice by the vaccination of KLH- conjugated Sph2(176-191), or the purified Sph2(176-191)-specific IgG from the sera of high responders by using affinity chromatography.
Some minor points:
1) Black background of Fig. 1 makes it not clear to read.
2) PDB ID should be provide for the Sph2 3D-structure (Fig. 1)
Author Response
Dear Reviewer,
Firstly, we want to thank you for your time and efforts in helping us to improve the manuscript quality. We looking to attending and revising every comment, and explaining here the modifications.
Best regards,
1) Would the authors add some information on the current development of leptospirosis vaccines, in the part of introduction? It would be helpful for the readers to understand the background and significance of their study.
Thank you for the suggestion. Based on this, we revised the introduction, inserting a topic of the current development of leptospirosis vaccines, as below.
“Despite the impact of leptospirosis on public and animal health, there is a lack of effective vaccines against the more than 300 antigenically diverse serovars of pathogenic Leptospira [1,10]. Until now, vaccines available against leptospirosis are either based on inactivated bacteria or membrane preparations from pathogenic Leptospira species, which do not provide cross-protection among the pathogenic Leptospira species and are associated with severe side effects [11]. Hence, molecules associated with Leptospira pathogenesis, virulence, infectivity, or survival have been investigated as both therapeutic targets and vaccine candidates.”
2) Would the authors provide more discussion on the leptospirosis patients who did not respond to Sph2(176-191) nor Sph2(446-459)? Did these patients respond to recombinant Sph2 protein? Considering the amino acid sequences of Sph2(176-191) in some Leptospira spp. are not conservative (Table 3), the authors should discuss the influence of the sequence of Sph2(176-191) on the serum responses in the patient population.
We agreed with this observation and revised the discussion. Regarding the reactivity to the recombinant protein, even in agreeing with this approach, we did not do this evaluation, based on recent articles that evaluated the reactivity to epitopes in all studied populations, not only in individuals who respond to recombinant protein. Moreover, we believe that the difference in reactivity also could be associated with differences in expression levels of Sph2 among patients, which could have an influence on both, the level of antibodies and the severity of symptoms.
3) The authors declared that clinical information implied the protective potential of Sph2(176-191) epitope. However, I can not read it from Fig. 3. The statistics on frequency of symptoms are quite controversial. The authors should revise the chapter 3.7 Associations between Humoral Response and Clinical Features to make it support their conclusion.
Thank you for this comment. You are absolutely right when you mentioned that the statistics of symptoms are controversial with the protective potential of antibodies against Sph2(176-191). However, due to our limited number of studied patients, and to the unknowledge of the real expression of Sph2 during their infections, would be unexpected to prove undoubtedly the protective potential of antibodies anti- Sph2(176-191). Despite this, in this study, we observed that only patients with no antibodies, or low level of antibodies (reactivity indexes minor than 2), against Sph2(176-191) presented symptoms of complications, like hemorrhage, while patients with high reactivity presented only mild symptoms. Because of this, we suggested this epitope as a sequence to be better explored in other studies.
Therefore, based on your comment, we revise chapter 3.7 to highlight this information and modified the discussion to improve the clarity of our data.
Moreover, I suggested the authors provide some direct evidence about the protective ability of the antibodies specific for Sph2(176-191) if possible. For example, they can test the antibodies induced in mice by the vaccination of KLH- conjugated Sph2(176-191), or the purified Sph2(176-191)-specific IgG from the sera of high responders by using affinity chromatography.
Thank you, we appreciated the suggestion. However, in this study, our main aim was to identify (in silico prediction and experimental validation) Sph2 B-cell linear epitopes, not necessarily protective epitopes. Considering this, we believe that the suggested experiments, despite being interesting and important, are not essential to our aims, especially if we consider that the most of studies that identify epitopes, only use in silico analysis, driving sequences to vaccine constructions without any experimental validation. In this context, we will not prove our hypothesis that epitope Sph2(176-191) could be a target of protective antibodies against leptospirosis. However, we believe that we will evaluate the protective potential and immunogenicity of this epitope, and others that are being identified by our group, soon, in a study of multi-epitope vaccines that we are developing in animal models.
Minor points:
1) Black background of Fig. 1 makes it not clear to read.
We modify the figure to a transparent background.
2) PDB ID should be provided for the Sph2 3D-structure (Fig. 1)
Thank you for the observation. The PDB ID was inserted into the paper.
Reviewer 3 Report
The manuscript identifies through bioinformatics tools 2 B-linear epitopes of sphingomyelinases from Leptospira sp and performs a broad serological correlation with the clinical status of the patients. The subject is important, and the work explores an important aspect of Leptospirosis differently. Therefore, the manuscript deserves to be published. However, it needs extensive correction and revision of some important elements.
Major points
1) Item 2.1-I needed help understanding Table 1 in the body of the text. It could be included as a Supplementary.
2) Item 2.4 and other parts of the text- Name of species must be written in italics.
3) As it is the first time these synthetic peptides are being described, the results (purification and masses) must be shown as supplementary, including the description of the gradients and the entire methodology.
4) Line 160, item 2.7 and other parts of the text - Describe the manufacturers' name, city, state, and country.
5) Information from lines 175-176 is repeated in lines 162-164.
6) Figure 3.1-Inform in the footer the meaning of the strong and weak marking of the letters of the amino acids of the epitopes.
7) 3.2. Inform from which SCORE value the algorithm considers the antigen as protective.
8) 3.4. –These studies deviate from the title if the epitopes involved are not part of the active center or cell binding.
9) Figure 1 is structurally uninformative and needs to present adequate resolution. There is no evidence of the secondary structure of the epitopes, and it does not adequately indicate the amino acids involved in the binding of the metal and host, nor the catalytic site. Some of this last information is described in the results, and it becomes difficult to visualize if they have some relationship with the binding site of the anti-peptide antibodies.
10) Line 264-265- “This statement regarding the three-dimensional location of the epitopes does not seem correct to me since no experiment or evidence is shown in work and cannot be based on the citation of another job. At most, it could be suggested that the binding of antibodies to epitopes could lead to steric hindrance, but the distance does not seem adequate to me. To prove this fact, further analyses by bioinformatics should be made to reach this conclusion.
11) Table 5- Inform in the caption or footer what RL and NRL mean
12) Inform in the text if only sera from a patient showing reactivity with IgM antibodies also show IgG reactivity for the same epitope.
13) Discussion-The presence of sphingomyelinase specifically in pathogenic Leptospires does not exclude the possibility that epitopes are present in other organisms and/or proteins from different organisms (cross-reactivity), even if the authors have consulted the Immune Epitope Database and PeptideAtlas. These repositories contain only the epitopes identified by any methodology but do not exclude the presence of the epitopes in other organisms and proteins. To corroborate this information, it is necessary to consult other sites that analyze the sequences of peptides in proteins.
14) Lines 416-420 repeated in work-Modify the sentences.
15) Line 104 informs the use of 9 algorithms, and in line 421, ten algorithms. Fix the information.
16) Abstract lines 32-34 and (conclusions) lines 551 and 553. This information should be modified or removed since, in work, there is no evidence of the robustness of the epitopes for use in diagnostic tests and/or vaccines. Therefore, studies do not support epitopes as candidates for developing a multi-epitope vaccine.
17) Correctly abbreviate multiple cited references
Author Response
Dear Reviewer,
Firstly, we want to thank you for your time and efforts in helping us to improve the manuscript quality. We looking to attending and revising every comment and critical, and to explain here the modifications.
Best regards,
1) Item 2.1-I needed help understanding Table 1 in the body of the text. It could be included as a Supplementary.
We understood your observation. However, we consider that this table summarizes general information on studied proteins. From our point of view, it allows a quick observation of proteins and pathogenic Leptospira spp. used in our study. Based on this, and in previous similar studies published by our group, and in the comments of the other two revisors, we maintained Table 1 in the body of the text.
2) Item 2.4 and other parts of the text- Name of species must be written in italics.
Thank you for the observation. All the text-Names of species were revised.
3) As it is the first time these synthetic peptides are being described, the results (purification and masses) must be shown as supplementary, including the description of the gradients and the entire methodology.
We appreciated the comment. Considering previous works that used the same approach, we revised the test and inserted references as below.
“Sequences predicted as antigenic linear B-cell epitopes were synthesized by fluorenylmethoxycarbonyl (F-moc) solid-phase chemistry [41,42] (WatsonBio, Houston, TX, USA). Analytical chromatography of the peptides demonstrated a purity degree higher than 95%, and mass spectrometry analysis of the peptides indicated estimated masses corresponding to the molecular masses of the peptides.”
4) Line 160, item 2.7 and other parts of the text - Describe the manufacturers' name, city, state, and country.
Thank you for the observation. We revised the manuscript to standardize manufacturers' information.
5) Information from lines 175-176 is repeated in lines 162-164.
To attend to this comment we removed the information in lines 162-164.
6) Figure 3.1-Inform in the footer the meaning of the strong and weak marking of the letters of the amino acids of the epitopes.
We believe that the comment was related to Table 2, in results 3.1. If it is the case, as previously indicated in Table Title, strongly marked letters mean amino acids predicted by each used algorithm. However, considering your comment we inform it in the footer as below.
“Black letters represent amino acids predicted by the algorithm and gray letters indicate non-predicted amino acids by each used algorithm.”
7) 3.2. Inform from which SCORE value the algorithm considers the antigen as protective.
The algorithm Vaxijen, classify proteins/peptides as “Protective antigen / Not protective antigen”, according to the selected threshold. In this study, we used the default threshold (0.4), as informed in 2.3 (between lines 145 and 148), as below.
“Each predicted epitope was evaluated for antigenicity by the VaxiJen algorithm (http://www.ddg-pharmfac.net/vaxijen/VaxiJen/VaxiJen.html), with the default threshold (0.4).”
8) 3.4. –These studies deviate from the title if the epitopes involved are not part of the active center or cell binding.
We are in disagreeing with this comment. Item 3.4 aimed to demonstrate the location of predicted epitopes in protein and to show the previously identified active center and cell-binding amino acids. Not necessarily, predicted epitopes need to be in active sites or cell binding regions to be an immunogenic, antigenic, or protective target. Moreover, here we compare the conservation of amino acids in the active center and cell-binding between several pathogenic Leptospira spp. (Table 4) and illustrated these amino acids and predicted epitopes in Figure 1. Based on this, we thank you for the observation but just revised the text to a better understanding.
9) Figure 1 is structurally uninformative and needs to present adequate resolution. There is no evidence of the secondary structure of the epitopes, and it does not adequately indicate the amino acids involved in the binding of the metal and host, nor the catalytic site. Some of this last information is described in the results, and it becomes difficult to visualize if they have some relationship with the binding site of the anti-peptide antibodies.
Based on your comment and suggestions from revisor 2, we modified the presentation of Figure 1. Now, in the cartoon, cylindrical helices, round helices, flat sheets, and smooth loops were applied to allow better visualization of Alpha-Fold predicted protein AF-P59116-F1(155-612). Regarding the resolution, now it is better than indicated in the author’s instructions, presented in 600 x 600 dpi, however, we believed that the resolution can be reduced in the manuscript.
10) Line 264-265- “This statement regarding the three-dimensional location of the epitopes does not seem correct to me since no experiment or evidence is shown in work and cannot be based on the citation of another job. At most, it could be suggested that the binding of antibodies to epitopes could lead to steric hindrance, but the distance does not seem adequate to me. To prove this fact, further analyses by bioinformatics should be made to reach this conclusion.
We understand your observation. Unfortunately, a better evaluation of active sites is based on crystallographic structures, that were unavailable to L. interrogans Sph2. Due to this, the alternative approach was the prediction of active sites, based on predicted 3D structures and other homologous proteins. Actually, we also did these analyses using bioinformatic approaches like Active Site Ligant (http://www.scfbio-iitd.res.in/dock/ActiveSite.jsp), Fpocket and Hpocket servers (https://bioserv.rpbs.univ-paris-diderot.fr/services/fpocket/), and others, however, we observed that using this approach, we would only reproduce data previously demonstrated by the study of Narayanavari and collaborators. Considering this, we chose to explore their data and compare the conservation of predicted sites between pathogenic Leptospira spp. Moreover, we used the Alphafold 3D predicted structure to highlight the localization of predicted epitopes. Regarding this, based on your comment, we revised the manuscript to improve the clearance of our text. Moreover, we agree with your conclusion and also inserted this idea in our discussion, reinforcing the necessity of novel studies to prove our hypothesis.
“However, studies aiming to identify the active sites of Sph2 from Leptospira spp., to evaluate their activity and the role of antibodies against their epitopes remain a lack of knowledge.”
11) Table 5- Inform in the caption or footer what RL and NRL mean
Thanks for the suggestion. We inform in the caption the meaning of RL and NRL as below.
RL: Reactive to Leptospira spp. (MAT positive), NRL: Non-reactive to Leptospira spp. (MAT negative).
12) Inform in the text if only sera from a patient showing reactivity with IgM antibodies also show IgG reactivity for the same epitope.
This information could be observed in Figures 2a and 2b, in which we showed the individual reactivity of IgM and IgG of each sample against each epitope (Figure 2a) and the frequency of IgM, IgG, and Overall (IgM + IgG) responders to each epitope (Figure 2b). However, based on your suggestion, we also inform it in the text, as below.
“Among responders against epitopes, only one individual exclusively presented IgM antibodies against Sph2(446-459), while other two IgM responders, also presented IgG antibodies.”
13) Discussion-The presence of sphingomyelinase specifically in pathogenic Leptospires does not exclude the possibility that epitopes are present in other organisms and/or proteins from different organisms (cross-reactivity), even if the authors have consulted the Immune Epitope Database and PeptideAtlas. These repositories contain only the epitopes identified by any methodology but do not exclude the presence of the epitopes in other organisms and proteins. To corroborate this information, it is necessary to consult other sites that analyze the sequences of peptides in proteins.
We agree with this comment. However, as informed on topic 3.3, we also evaluated the similarity of predicted epitopes with other sequences by BlastP (“Regarding the conservation degree of predicted epitopes, the epitopes Sph2(176-191) and Sph2(446-459) were considered non-conserved among microorganisms not belonging to the genus Leptospira, because presented an E-value greater than 1 on BlastP.”). Nevertheless, based on your comment we revised the discussion to highlight this data.
14) Lines 416-420 repeated in work-Modify the sentences.
Thank you for the observation. We revised the text in the manuscript.
15) Line 104 informs the use of 9 algorithms, and in line 421, ten algorithms. Fix the information.
We appreciated the observation and revised the text. Both information, line 104 and line 426, were right since the first information is related to the prediction of B-cell linear epitopes (using nine algorithms), which was further analyzed by one another algorithm to predict their antigenicity (totalizing 10 algorithms used to predict antigenic and linear B-cell epitopes). However, considering the comment, we revised the text as below.
“Firstly, we used a combination of algorithms to predict linear B-cell epitopes…”
16) Abstract lines 32-34 and (conclusions) lines 551 and 553. This information should be modified or removed since, in work, there is no evidence of the robustness of the epitopes for use in diagnostic tests and/or vaccines. Therefore, studies do not support epitopes as candidates for developing a multi-epitope vaccine.
We agree with your comment and revised the text. However, we highlight that the most of designed multi-epitope vaccines are based only in silico predicted epitopes, without any experimental validation of antigenicity, immunogenicity, or protection.
17) Correctly abbreviate multiple cited references
References were done using the program EndNote with the MDPI pattern. Nevertheless, we revised all references to correct any failures.
Round 2
Reviewer 3 Report
All suggestions were properly accepted and/or answered, except for question 13, in which in most cases using only Blast it is not possible to identify small sequences (epitopes). This finer analysis necessitates the use of other approaches. However, as this information does not affect the set and final result, we considered the manuscript accepted for publication, in its present form.